# A quasi-integral controller for adaptation of genetic modules to variable ribosome demand

Hsin-Ho Huang[1], Yili Qian [1] & Domitilla Del Vecchio[1]

The behavior of genetic circuits is often poorly predictable. A gene's expression level is not only determined by the intended regulators, but also affected by changes in ribosome availability imparted by expression of other genes. Here we design a quasi-integral biomolecular feedback controller that enables the expression level of any gene of interest (GOI) to adapt to changes in available ribosomes. The feedback is implemented through a synthetic small RNA (sRNA) that silences the GOI's mRNA, and uses orthogonal extracytoplasmic function (ECF) sigma factor to sense the GOI's translation and to actuate sRNA transcription. Without the controller, the expression level of the GOI is reduced by 50% when a resource competitor is activated. With the controller, by contrast, gene expression level is practically unaffected by the competitor. This feedback controller allows adaptation of genetic modules to variable ribosome demand and thus aids modular construction of complicated circuits.

[1] Massachusetts Institute of Technology, Department of Mechanical Engineering, 77 Massachusetts Avenue, Cambridge, MA 02139, USA. These authors contributed equally: Hsin-Ho Huang, Yili Qian. Correspondence and requests for materials should be addressed to D.D.V. (email: ddv@mit.edu)

The ability to create complicated systems from composition of functional modules is critical to the progress of synthetic biology, yet it has been a longstanding challenge in the field[1–3]. Although progress has been made toward this goal[2–6], context-dependent behavior of genetic circuits is still a major hurdle to modular design[1,3]. This frequently leads to a combinatorial design problem where one has to redesign a circuit's components when a new component is added. While many factors contribute to context-dependence of genetic modules, sharing limited gene expression resources is a major player in this problem[7–13].

In bacteria, the rate of gene expression is mainly limited by the availability of ribosomes[7,8,13]. In particular, activation of one transcriptional (TX) device, that is, a system where input TX regulators affect expression level of one output protein, reduces availability of ribosomes to other, otherwise unconnected, TX devices, affecting their gene-expression levels by up to 60%[7,11]. These unintended nonregulatory interactions among TX devices can significantly alter the emergent behavior of genetic circuits. For example, the dose–response curve of a genetic activation cascade can be biphasic or even monotonically decreasing, instead of being monotonically increasing as expected from the composition of its TX devices[9]. Therefore, there is a general need to find solutions that make the expression level of a TX device adapt to changes in ribosome availability while keeping the input and output connectivity of the device unchanged. This would enable seamless and scalable composition of TX devices whose input/output (i/o) behaviors are more robust to context.

In traditional engineering systems, feedback control has played a critical role in making circuit's components modular, i.e., in maintaining a desired i/o response despite disturbances. This enabled predictable and reliable composition of larger systems from subsystems[14]. In synthetic biology, negative feedback control has been employed, for example, to reduce gene-expression heterogeneity[15–19], to speed up the response of gene transcription networks[20], to maintain cell growth in the presence of cellular burden[21], and to optimize output from a TX device with activator overdosage[22]. Recently, feedback control has been considered to increase robustness of gene expression to fluctuations in available resources. In particular, Hamadeh et al.[23] carried out theoretical analysis to compare the performance limits of different feedback architectures to mitigate effects of resource competition at various levels of gene expression. In the case where the major resource competed for is ribosomes, a post-TX controller is theoretically sufficient for robustifying gene expression in the face of resource fluctuations. Shopera et al.[24] implemented a TX controller by co-designing the regulatory input and output protein of a TX device such that the input can be sequestered by the anti-activator output protein, creating a negative feedback loop. As a result, the expression level of the TX device's gene is more robust to fluctuations in availability of ribosomes. Although a promising proof of concept, the requirement to co-design the TX device's input and output to engineer the feedback prevents generalization and scalability of this solution beyond the specific circuit's instance considered.

In this paper, we design a post-TX feedback controller in which alteration of a TX device's intended regulatory input and output is not necessary to engineer adaptation to changes in ribosome availability. Specifically, we design and implement a biomolecular feedback controller mediated through synthetic small RNA (sRNA)[25] to enable adaptation of the TX device's output protein concentration to changes in ribosome availability. The key innovation of our solution is a quasi-integral control (QIC) strategy, which can approximate ideal integral control when all reactions constituting the controller are sufficiently faster than molecular decay[26]. Integral controllers are often responsible for homeostasis and perfect adaptation in natural systems[27,28] and are ubiquitous in traditional engineering systems to ensure robustness to disturbances[29]. While genetic circuit motifs implementing integral control have been proposed[30], they may be difficult to realize due to decay (i.e., dilution and degradation) of the biomolecules implementing the control reactions, which leads to integrator leakiness[26,31,32]. A QIC manages this physical constraint by implementing all control reactions with sufficiently large reaction rates that are tunable through a feedback gain, mitigating the effects of integration leakiness. Based on this design principle and mathematical analysis of the circuit's model, we construct and test a library of regulated TX devices with variable feedback gains. As predicted from the model, while the output of an unregulated TX device is reduced by 50% when a resource competitor is activated, a high-gain regulated TX device with similar expression level keeps its output nearly unchanged when the same resource competitor is activated.

## Results

**Enforcing modularity through embedded feedback control**. A genetic circuit is commonly viewed as the composition of TX devices, which are systems that take TX regulators as inputs and produce proteins (possibly also TX regulators) as outputs. Modularity, the property according to which the i/o behavior of a system does not change with its context, is required for bottom-up design of synthetic genetic circuits[33]. However, the salient properties of a TX device often depend on its context, which includes both direct connectivity to and mere presence of other TX devices[1,3]. While problems of loads due to direct connectivity have been addressed in earlier works[4,5], the problem of context-dependence due to resource sharing remains largely open[7,9–12]. Figure 1 shows a cartoon of this problem along with the solution that we aim to implement in this paper. Specifically, with reference to Fig. 1a, the demand for ribosomes imparted by one TX device (TX device 2) in response to its regulatory input ($u_2$) leads to a change $d$ in the concentration of available ribosomes. This, in turn, affects the output ($y_1$) of a different TX device (TX device 1), despite its intended regulatory input ($u_1$) is unchanged (Fig. 1b). As a result, the two TX devices become indirectly connected, breaking modularity, and confounding circuit design.

To restore modularity of TX device 1, we seek to design a post-TX feedback controller embedded in this device with the aim of making the device's i/o behavior independent of TX device 2. To compensate for a change in the device's output $y_1$ imparted by a drop or an increase $d$ in free ribosome concentration, this post-TX controller senses the TX device 1's translation (TL) rate, which is proportional to free ribosome concentration, and then appropriately adjust the device's mRNA level $m_1$ through post-TX mRNA processing (Fig. 1c). As a result, under a fixed input $u_1$, the output $y_1$ of TX device 1 should ideally adapt to changes in ribosome demand by TX device 2 (Fig. 1d). Since this feedback design is post-TX, it is orthogonal to the intended regulatory input $u_1$ and output $y_1$ of TX device 1, enabling seamless composition of multiple TX devices through pre-defined TX regulatory interactions.

The problem of making the output of TX device 1 independent of TX device 2's increased demand for ribosomes can be solved by regarding the variation in ribosome availability $d$ as a disturbance to TX device 1 and solving the control theoretic problem of disturbance rejection[29]. Rejection of (i.e., perfect adaptation to) a disturbance can be accomplished by integral feedback control. An ideal integral feedback controller computes the difference between the output of interest $y$ and the desired output $\bar{y}$ as the error $e(t) = \bar{y} - y(t)$. A memory element $z$ in the controller then

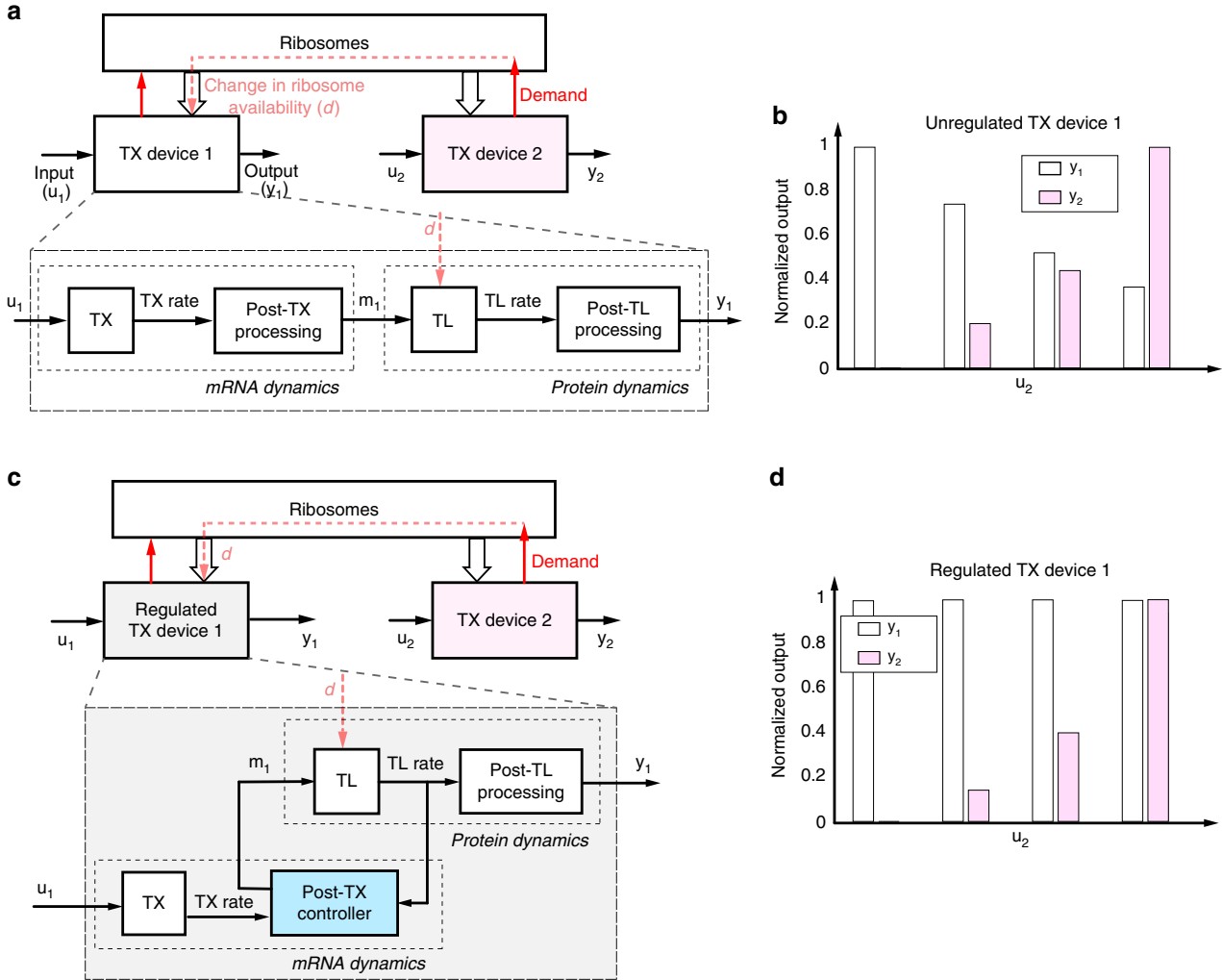

**Fig. 1** The problem of modularity in resource-limited genetic circuits. **a** Each TX device $i$ ($i = 1, 2$) takes a TX regulator as input ($u_i$) and produces a protein as output ($y_i$). Each device can be decomposed into the cascade of key biological processes, here depicted as transcription (TX), post-TX mRNA processing, translation (TL), and post-TL protein processing. The TX (TL) block takes as input a TX regulator's (mRNA) concentration and gives as output the TX (TL) rate. The post-TX (TL) mRNA (protein) processing block takes TX (TL) rate as input and produces mRNA (protein) concentration as output. These blocks can include processes affecting the stability of molecules (including dilution due to cell growth) or molecule activity (i.e., covalent modification). When multiple TX devices share a pool of limited ribosomes, the demand from one device causes a change $d$ in ribosome availability, which affects TL in other devices, creating unintended interactions among TX devices. **b** Example behavior for the two-device circuit depicted in (**a**). The output from TX device 1 ($y_1$) becomes coupled to the input to TX device 2 ($u_2$), resulting in a circuit that lacks modularity. **c** Block diagram of an embedded post-TX feedback control design. A post-TX controller senses the TL rate of the GOI, takes as input the TX rate of the GOI, and, based on these two quantities, adjusts the mRNA level by post-TX mRNA processing in order to compensate for the effect of $d$. **d** The desired outcome from a regulated TX device is that despite TX device 2 is increasingly activated, the output of TX device 1 remains unchanged when presented with a constant input $u_1$

accumulates (i.e., integrates) this error over time according to: $z(t) = \int_0^t e(\tau)\mathrm{d}\tau$. The memory of the error $z$ is then used to determine the amount of actuation (in our case, the mRNA level $m_1$, see Fig. 1c) applied to the process to be regulated (i.e., TL, see Fig. 1c). If the feedback interconnection of the process to be regulated and the integral controller is stable, then, at steady state, we have that $\mathrm{d}z/\mathrm{d}t = 0$, which leads to $e = 0$ regardless of the disturbance $d$. Due to this property, integral control is applied ubiquitously to engineering systems (e.g., vehicle cruise control systems[29]) and has been identified in mathematical models of natural networks (e.g., bacterial chemotaxis[27] and calcium homeostasis[28]). Therefore, with reference to Fig. 1c, we aim to design a post-TX controller that implements integral-control-like behavior to render the output of TX device 1 independent of TX device 2.

**Design of an embedded post-TX controller via sRNA silencing**. Figure 2a illustrates the genetic circuit diagram of the specific post-TX controller that we design in this paper. In order to evaluate the ability of a post-TX controller to make the output of TX device 1 adapt to changing ribosome demand by TX device 2, we assembled a library of test-bed genetic circuits with two TX devices shown in Fig. 2a. Specifically, since TX device 2 needs to apply a variable demand for ribosomes, we made TX device 2 externally inducible and made red fluorescent protein (RFP) as its output to assess ribosome demand. We embedded a post-TX controller enabled by sRNA silencing in TX device 1. For this device, we chose constitutive promoters since our design needs to demonstrate that the TX device's output (green fluorescent protein (GFP) level) stays unchanged when its input is kept constant, despite a change in ribosome availability. This is a model system

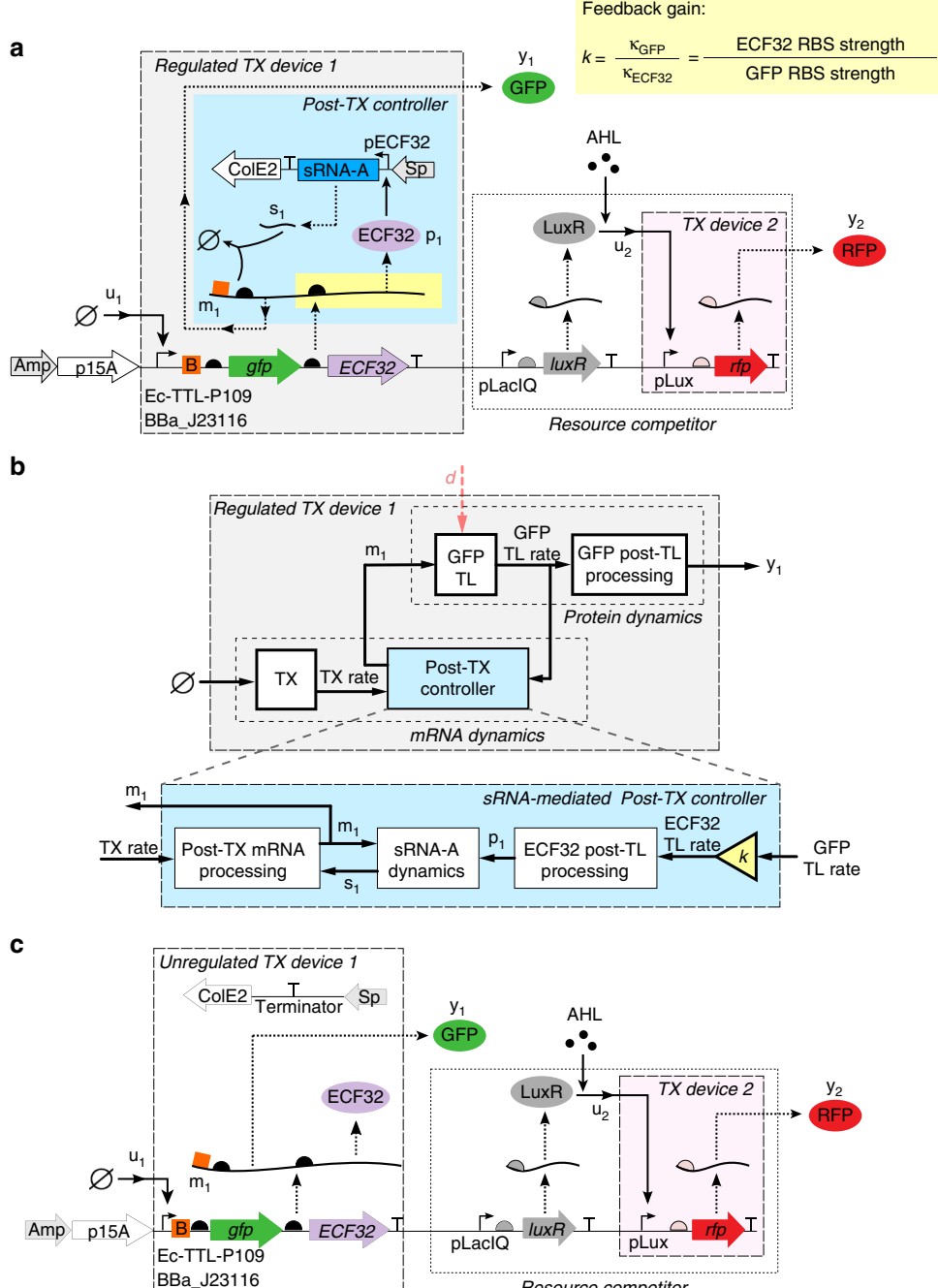

**Fig. 2** sRNA-mediated post-TX controllers. **a** Circuit diagram of a regulated TX device 1 with inducible TX device 2 functioning as a resource competitor. Upwards arrow with tip rightwards indicate promoters, semicircles represent RBSs, orange box with letter B stands for sRNA-A targeting sequence, and "T" symbols represent terminators. The feedback controller consists of ECF32, co-transcribed with GFP, which is used to sense GFP translation rate and to actuate transcription of sRNA-A. sRNA-A antisenses its targeting sequence on the mRNA for degradation of both RNA molecules. **b** Block diagram representation of the sRNA-mediated post-TX controller. The ratio between ECF32 and GFP TL rates is $k$ as defined in panel **a** (see Supplementary Eq. (9) and Supplementary Note 7 for derivation), which we call here the feedback gain. ECF32 TL rate enters the ECF32 post-TL processing block, which produces ECF32 protein ($p_1$) as output. This, in turn, enters as an input the process that produces sRNA-A $s_1$ (sRNA-A dynamics), which is constituted by sRNA-A TX through pECF32 and sRNA degradation (mRNA coupled and uncoupled). As a last step, sRNA-A enters the mRNA $m_1$ post-TX processing block to silence mRNA $m_1$. **c** Circuit diagram of an unregulated TX device 1 with inducible TX device 2 functioning as a resource competitor. The pECF32 promoter and sRNA-A in (**a**) are removed

for studying competition for ribosomes as employed in earlier works[7,11,24,34]. Specifically, referring to Fig. 2a, a pLacIQ promoter is used to constitutively express LuxR. In the presence of LuxR's effector $N$-hexanoyl-L-homoserine lactone (AHL), AHL-bound LuxR ($u_2$) is formed to activate TX device 2 to produce disturbance output RFP ($y_2$). These two genes constitute a

resource competitor that demands more ribosomes when AHL concentration increases, leading to a decrease in the amount of available ribosomes to translate mRNAs in TX device 1.

TX device 1 embeds the sRNA-mediated post-TX controller. The controller consists of four key biological parts: sRNA-A and its targeting sequence[25], extracytoplasmic function (ECF) sigma

factor 32_1122 (referred to as ECF32 hereafter), and its cognate promoter pECF32[35]. The choice of ECF32 and pECF32 is based on their minimal impact on host cell growth and their dose response's wide dynamic range (Supplementary Note 1). Specifically, *ECF32* gene is introduced downstream of the output *gfp* gene to form a bi-cistronic operon. Co-transcription of *gfp* and *ECF32* genes is driven by a constitutive promoter (Fig. 2a). We constructed circuits that either employ the stronger constitutive promoter Ec-TTL-P109[36] or the weaker constitutive BioBrick promoter BBa_J23116. This provides a means to adjust the output level $y_1$ of TX device 1 when comparing the performance of different TX devices. The mRNA co-transcript of *gfp* and *ECF32* genes has the sRNA-A's targeting sequence immediately upstream of the *gfp* gene's ribosome binding site (RBS). The sRNA can complementarily pair with its target mRNA, forming an inert RNA complex that is rapidly cleaved by RNase E, leading to coupled degradation of both mRNA and sRNA[37–41].

Since GFP and ECF32 proteins are translated from the same mRNA co-transcript $m_1$ using the same pool of ribosomes, this bi-cistronic operon design allows ECF32 TL rate to be proportional to GFP TL rate by a factor $k$, which is the ratio between GFP and ECF32 RBS dissociation constants (see Supplementary Eqs. (8) and (9) for derivation):

$$k = \frac{\kappa_{GFP}}{\kappa_{ECF32}}. \qquad (1)$$

This relationship allows the sRNA-mediated post-TX controller to sense a change in GFP TL rate (input to the controller) through ECF32 TL rate and respond to this change by adjusting subsequent ECF32 protein and sRNA concentrations, which subsequently determine mRNA co-transcript concentration $m_1$ as the controller's output. More specifically, with reference to the block diagram in Fig. 2b, when the resource competitor is activated, the amount of ribosome available to translate GFP decreases, causing a reduction $\Delta$ in its TL rate. Due to the bi-cistronic operon design, a decrease in GFP TL rate is always accompanied by a $k\Delta$ decrease in ECF32 TL rate. Reduction in ECF32 TL rate, in turn, leads to a decreased ECF32 concentration $p_1$, which decreases sRNA-A's TX, leading to an increase in the concentration of GFP/ECF32 mRNA co-transcripts $m_1$. This allows GFP TL rate to recover, closing the feedback loop. Based on the above reasoning, a regulated TX device with a higher $k$ responds to the same decrease $\Delta$ in GFP TL rate with a larger decrease in sRNA transcription, leading to increased control action (sRNA-A transcription). Therefore, we call $k$, as defined in Eq. (1), the feedback gain of the sRNA-mediated embedded post-TX controller.

To experimentally evaluate the benefit of the sRNA-mediated post-TX controller, we constructed another library of circuits with unregulated TX device 1 and same resource competitor (Fig. 2c). In particular, the only difference is the absence of pECF32 promoter and sRNA-A message in the unregulated device. As a result, the mRNA level in the unregulated TX device 1 ($m_1$) is not responsive to the output $y_1$ TL rate, breaking the feedback loop.

**The sRNA-mediated feedback implements quasi-integral control**. If we assume that (A) GFP and ECF32 proteins decay with the same rate constant $\gamma$ and that (B) we start our experiment from steady state gene expression, then our circuit design allows ECF32 protein concentration to be theoretically proportional to GFP concentration by the feedback gain $k$: $p_1(t) = ky_1(t)$ (see Supplementary Eqs. (13) and (14)). As a result, the biomolecular reactions constituting the regulated TX device 1 (Fig. 2a) can be

described by the following ordinary differential equations (ODEs):

$$\frac{d}{dt}m_1 = TDH(u_1) - \lambda m_1 s_1/\beta - \delta m_1, \qquad (2)$$

$$\frac{d}{dt}s_1 = kT_s y_1 - \lambda m_1 s_1/\beta - \delta s_1, \qquad (3)$$

$$\frac{d}{dt}y_1 = R(1-d)m_1/\kappa_{GFP} - \gamma y_1, \qquad (4)$$

where $m_1$, $s_1$, and $y_1$ stand for the concentrations of the GFP/ECF32 mRNA co-transcript, sRNA, and GFP protein, respectively. In this model, function $H(u_1) \in [0, 1]$ describes TX regulation by TF input $u_1$ and $H(u_1) \equiv 1$ when the TX device's promoter is constitutive. Parameter $D$ is the plasmid copy number of the regulated gene; $T$ is the TX rate constant per DNA copy, which is primarily dictated by the TX device's promoter strength; $T_s$ is a lumped TX rate constant for sRNA, which is proportional to its plasmid copy number and pECF32's promoter strength; $k$ is the feedback gain defined previously in Eq. (1); $\delta$ is the decay rate constant of uncoupled mRNA and sRNA, which we assume to be identical for both species for simplicity (see Supplementary Note 7 for full analysis without this assumption); $\lambda$ is the mRNA–sRNA decay rate constant; $\beta$ is the dissociation constant of mRNA–sRNA binding, and $R$ is the maximum TL rate constant proportional to the total amount of ribosomes. The disturbance input $0 \le d < 1$ models the fold change in free ribosome availability due to competitor activation. It is 0 when free ribosome concentration is unchanged and it approaches 1 when there is almost no free ribosome available. The models of RNA transcription, decay and RNA–RNA interaction in Eqs. (2) and (3) are standard and can be found in refs. [37,42–44]. Nevertheless, we derived them from chemical reactions in Supplementary Note 7.

If the uncoupled RNAs do not decay (i.e., $\delta = 0$), this sRNA-mediated feedback system is an antithetic integral controller[30], which performs ideal integral feedback control. Specifically, the integral action in the RNA controller dynamics (2) and (3) appears through the memory variable $z := m_1 - s_1$, whose dynamics follow

$$\frac{d}{dt}z = TDH(u_1) - kT_s y_1 = kT_s e, \Rightarrow z(t) = kT_s \int_0^t e(\tau)d\tau, \quad (5)$$

where, we have defined $e(t) = TDH(u_1)/(kT_s) - y_1(t)$. Suppose that the ODE model (2) and (4) is stable, then the time derivative of $z$ in (5) reaches 0 at steady state, resulting in the steady state output to be $y_1 = TDH(u_1)/(kT_s)$, which is independent of the disturbance $d$. This implies that expression of the regulated TX device 1 ($y_1$) is independent of (i.e., adapts perfectly to) the disturbance $d$ and only depends on its own regulatory input $u_1$. However, due to (i) degradation by RNase and (ii) dilution due to fast growth of bacteria, decay of uncoupled mRNA and sRNA are unavoidable in practice[37] ($\delta > 0$). As a consequence, the memory variable dynamics become

$$\frac{d}{dt}z = kT_s e - \delta z, \qquad (6)$$

which, in contrast to Eq. (5), is not an integral of the error. As a consequence, the memory $z$ gradually fades away due to decay of RNA species. This disruption of the key integral control structure can result in potentially large adaptation error, making it practically impossible for the output $y_1$ to adapt perfectly to a variation $d$ in ribosome availability.

Nevertheless, near perfect adaptation can be achieved by this sRNA-mediated post-TX controller by rational choice of the circuit parameters. Qualitatively, to overcome the undesirable effect of fading memory, one can engineer the accumulation of the error to take place at a much faster rate than the rate at which uncoupled mRNA and sRNA decay. This allows the memory signal to be amplified, making its decay negligible, and thus leading to practically no adaptation error. This strategy is called quasi-integral control and a more in-depth mathematical analysis of its properties can be found in[26]. For the sRNA-mediated controller described by Eqs. (2)–(4), the accumulation of the error can take place at a much faster rate than the rate of uncoupled mRNA and sRNA decay if all controller reactions (mRNA–sRNA interaction, mRNA transcription, and sRNA transcription) are much faster than decay rates. Specifically, this can be achieved (see Supplementary Note 8 for details) by satisfying the following parameter relationships: (I) $\delta/\lambda \ll 1$, (II) $\delta/T \ll 1$, and (III) $\delta/(kT_s) \ll 1$. In practice, condition (I) is readily satisfied due to the fact that sRNA-enabled mRNA degradation is much faster than degradation of mRNA by RNase[37–41] and condition (II) can be easily satisfied for regulated TX devices driven by a reasonably strong constitutive promoter, including the ones we use here (see analysis in Supplementary Note 8). Condition (III) can be reached by either having a sufficiently large $T_s$, which is dictated by the pECF32 promoter activity, or by a large feedback gain $k$. Since the pECF32 promoter is not particularly strong (Supplementary Fig. 5), condition (III) can be more easily achieved by increasing the gain $k$ as desired, which can be accomplished by simply increasing the ECF32 RBS strength. Hence, the feedback gain $k$, which can be tuned through the ECF32 RBS strength, dictates the ability of the controller to adapt to changes in available ribosomes.

These model predictions are confirmed through simulations in Fig. 3a. While the output of the unregulated TX device 1 changes significantly when ribosome availability drops, the output of the regulated TX device 1 with a high feedback gain (i.e., large $k$) adapts near perfectly to changes in ribosome availability $d$. By contrast, a regulated TX device 1 with a low-feedback gain (i.e., small $k$) behaves similarly to an unregulated device. In Fig. 3b, we also simulated the temporal response of a high-gain regulated TX device subject to a step decrease in ribosome availability. Due to the fast reactions constituting the controller, the step disturbance does not lead to prolonged transient dynamics. These simulation results confirm that we can construct a modular TX device by embedding in it an sRNA-mediated controller with high feedback gain, which can be tuned by ECF32's RBS strength.

**Experimental confirmation of the sRNA-A silencing mechanism**. Before testing the circuits in Fig. 2a, c, we first verified that activation of ECF32 can lead to enhanced sRNA-A transcription and subsequent inhibition of GFP expression. To this end, we performed a silencing experiment using the circuit in Fig. 4a (see plasmid maps in Supplementary Fig. 2). Specifically, the mRNA of *gfp* gene to be silenced has sRNA-A's targeting sequence (orange B box) located immediately upstream of the gene's RBS and it is transcribed by a strong constitutive promoter BBa_J23100 from the BioBrick Registry. sRNA-A is transcribed by pECF32 promoter in the presence of ECF32, whose expression is regulated by TetR repressor and its effector anhydrotetracycline (aTc). We confirmed that increasing the concentration of aTc decreases GFP expression, resulting in complete gene silencing of GFP expression at high aTc concentrations. In contrast, when pECF32 promoter and sRNA-A message were removed, we observed high levels of GFP expression that is independent of aTc

concentration (Fig. 4b). These results confirm that the sRNA-A can silence GFP expression upon ECF32 activation as desired.

**Tuning regulated device performance via the feedback gain**. Here, we experimentally demonstrate that, as predicted from the mathematical analysis section, increasing the feedback gain $k$ through the ECF32 RBS strength is an effective way to achieve adaptation to changing ribosome demand. We constructed and tested the circuits in Fig. 2a (c) consisting of a resource competitor and a regulated (unregulated) TX device. The plasmid maps and DNA sequences can be found in Supplementary Fig. 3 and Supplementary Note 2, respectively. To assess the performance of each regulated TX device from experimental data, we use its robustness as our main performance metric. In particular, robustness of a TX device is defined as the percentage of GFP expression when the resource competitor is fully activated (i.e., AHL = 1000 nM) relative to its nominal output, which is the GFP expression when the resource competitor is inactive (i.e., AHL = 0):

$$\text{Robustness} = \frac{\text{GFP expression when AHL} = 1000\,\text{nM}}{\text{GFP expression when AHL} = 0} \times 100\%. \tag{7}$$

Based on this performance metric, robustness of a modular TX device is 100%. From our analysis and the simulation in Fig. 3a, the ability of the post-TX controller to adapt to variable ribosome availability is dictated by a high feedback gain $k$, which can be increased in our circuit by increasing the RBS strength of ECF32. Therefore, we constructed a library of six regulated TX devices with three different ECF32 RBSs and two different promoters (Supplementary Table 3). In particular, ECF32 RBSs have TL initiation rates (TIRs) of 565, 1127, and 6474, as calculated by the RBS calculator 2.0[45], allowing the regulated TX devices to be equipped with low, medium and high gains, respectively. The RBS of GFP is kept unchanged with a TIR of 974 in all experiments. The promoter of a regulated device is either Ec-TTL-P109 (stronger)[36] or BBa_J23116 (weaker). Assessing the performance improvement with increased feedback gain under different promoter strengths allows us to validate that the design principle is independent of GFP level.

We first evaluated the robustness and nominal output for all six regulated TX devices (Fig. 5a). We find that, independent of the choice of promoter, increasing feedback gain $k$ promotes robustness. Specifically, for the circuits with the stronger (weaker) promoter, increasing feedback gain from low-to-high improved robustness from about 60% (50%) to about 90% (75%). As predicted by our model, the regulated devices employing high feedback gain are most robust to resource competitor activation (Fig. 5a). By contrast, the low-gain regulated devices are the least robust.

We then compared the performance of the high-gain and the low-gain regulated TX devices for the same promoter (Fig. 5b). While both the regulated and unregulated TX devices with low gain suffered more than 40% decrease in GFP level when the resource competitor was maximally activated, the high-gain regulated TX device was practically unaffected by activation of the resource competitor (Fig. 5b). Since both the regulated and the unregulated devices contain the *ECF32* gene (Fig. 2a, c), these results confirm that the robustness of the high-gain regulated device is not due to the mere presence of ECF32. Furthermore, RFP expression levels are comparable in all experiments (Fig. 5b and Fig. S10), indicating comparable disturbances created by the competitor on all circuits. Growth rates of cells containing unregulated high-gain TX device were appreciably slower than those of cells bearing high-gain regulated TX devices. This is

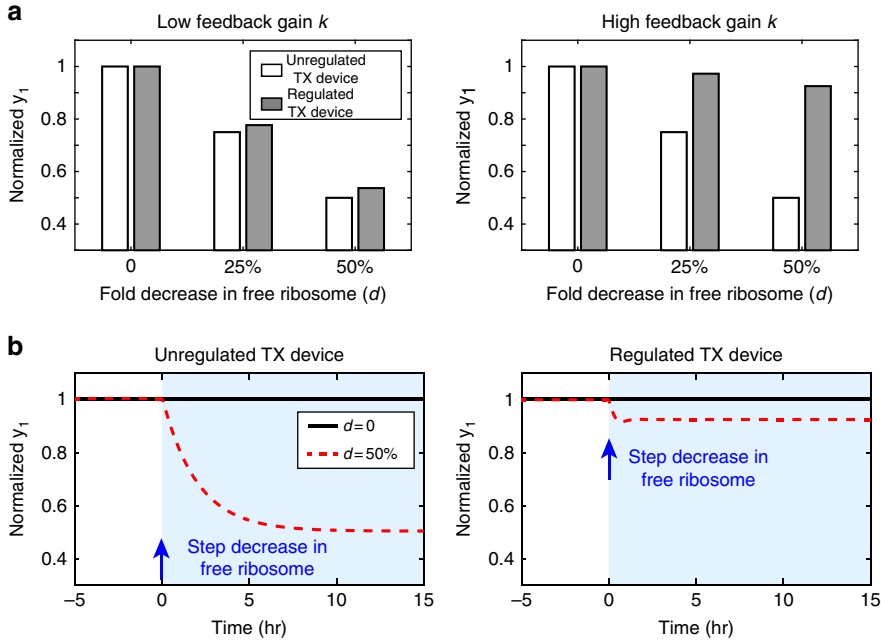

**Fig. 3** A regulated TX device with high feedback gain can reach almost perfect adaptation. **a** Simulation results of the steady states of Eqs. (2)–(4) with different disturbance values d. **b** Simulation of the temporal response of the GFP output $y_1$ from the unregulated (circuit in Fig. 2c) and regulated (circuit in Fig. 2a) TX devices. A step decrease in free ribosome concentration is introduced at $t = 0$ with the value of d indicated in the legend. Simulation parameters are listed in Supplementary Table 9. Source data are provided as a Source Data file.Source Data

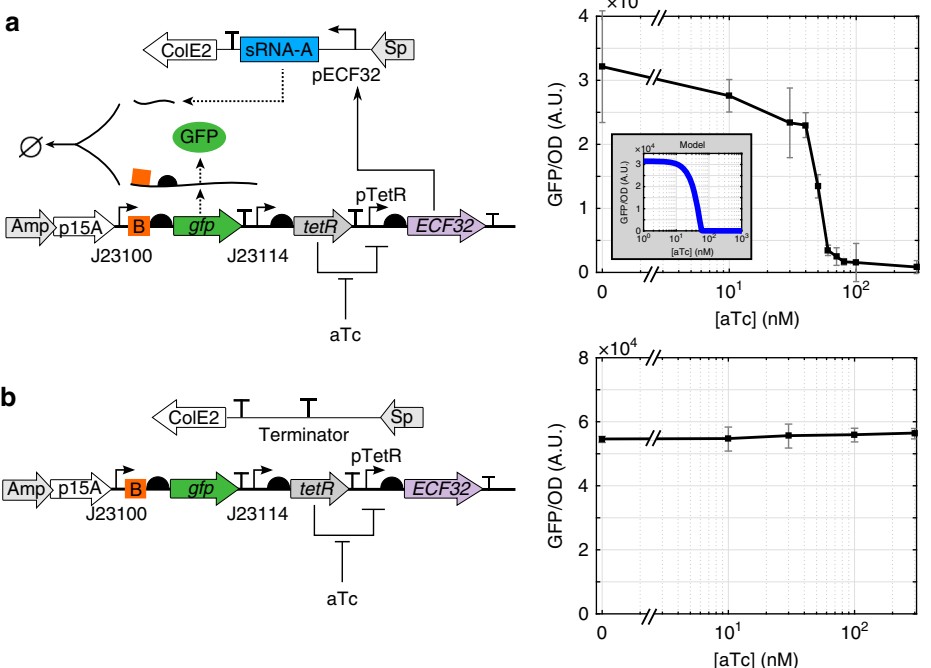

**Fig. 4** Confirmation of the sRNA-A silencing and the ECF32 actuating mechanisms. **a** GFP expression is regulated by the sRNA-A silencing and the ECF32 actuating mechanisms. When TetR-regulated ECF32 expression is induced by TetR's effector aTc, ECF32 binds to its cognate promoter pECF32 to transcribe sRNA-A. sRNA-A then antisenses to its targeting sequence (orange B box) on GFP's mRNA to silence GFP expression. Experimental results match well with prediction from the sRNA silencing model in Supplementary Note 10 and Supplementary Fig. 15 (shown in the gray inset box). For [aTc] = 0, 10, 30, 100, 300 nM, data represent the mean (±standard deviation) of eight replicates, including two biological replicates each with four technical replicates. For the rest of the aTc concentrations, data represent the mean (±standard deviation) of four replicates, including two biological replicates each with two technical replicates. **b** When pECF32 promoter and sRNA-A message were removed, GFP expression was not affected by aTc induction. Data were acquired with microplate photometer. GFP per OD values in arbitrary unit (A.U.) at least 8 h after induction were used as steady states to obtain the dose response curves in both panels. The error bars indicate standard deviation of four replicates, including two biological replicates, each with two technical replicates. See Supplementary Fig. 6 for complete experimental temporal responses of the circuit in panel **a**. Source data are provided as a Source Data file.Source Data

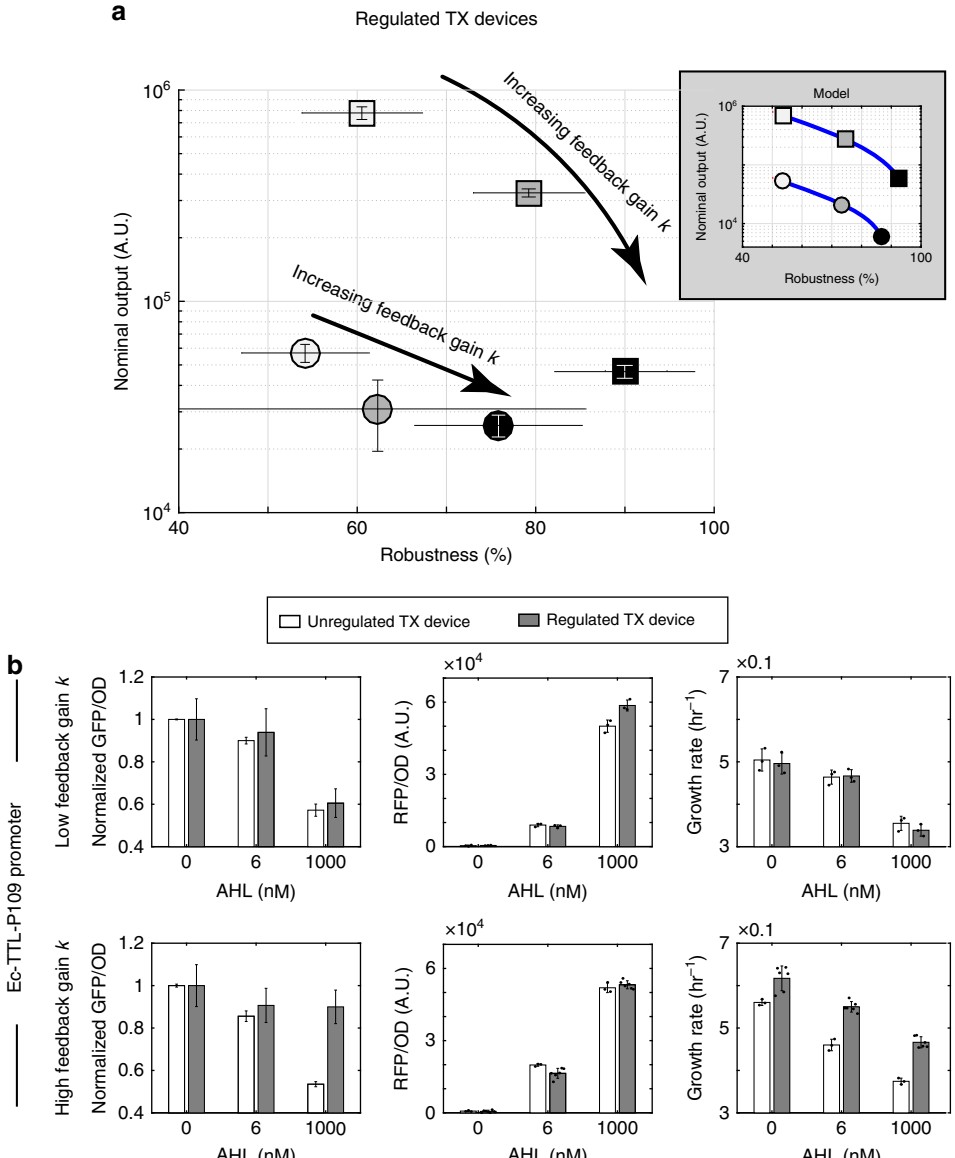

**Fig. 5** Performance of a regulated TX device is determined by its feedback gain. **a** Increasing feedback gain enhances the robustness of a regulated TX device driven by either the stronger Ec-TTL-P109 promoter (a square symbol) or the weaker BBa_J23116 promoter (a circle symbol). A symbol's color filling in gray scale from light to dark represents a feedback gain from low to high. Simulation results for model (2)–(4) with parameters in Supplementary Table 9 are in the inset box (see Supplementary Fig. 18 for additional simulation results). **b** GFP outputs, growth rates, and RFP outputs for low-gain and high-gain regulated and unregulated TX devices using the Ec-TTL-P109 promoter. GFP per OD values in arbitrary unit (A.U.) were normalized to their respective nominal outputs. All experimental data were obtained with a microplate photometer. Data with error bars represent mean values ± standard deviations. The regulated TX device using Ec-TLL-P109 promoter and high feedback gain has six replicates (three biological replicates each with two technical replicates). Other TX devices have three biological replicates. Specific value of an independent experiment is presented as a black dot. See Supplementary Fig. 10 for data for TX devices using BBa_J23116 promoter and/or medium feedback gain. Supplementary Figs. 7 and 8 contain complete temporal responses of all 12 circuits. Source data are provided as a Source Data file.Source Data

because nominal output from the unregulated TX device was higher than the output of its regulated counterpart, imposing a larger load on cell growth (Fig. 5b). We observed similar trends when comparing the medium-gain regulated devices with their unregulated counterparts and their detailed temporal and dose responses can be found in Supplementary Figs. 7, 8, and 10.

**Regulated and unregulated devices with similar output levels.** In order to further confirm that the controller performance is independent of GFP level, we performed a detailed comparison

between an unregulated device and a regulated device with high-gain and with comparable nominal output. These two devices are represented by the white circle and black square, respectively, in Fig. 6a. The regulated TX device uses the stronger Ec-TTL-P109 promoter and is embedded with an sRNA-mediated post-TX controller with ECF32 whose RBS TIR is 6474 (i.e., high feedback gain), while the unregulated TX device is only different in that it uses the weaker BBa_J23116 promoter and removes pECF32 promoter and sRNA-A message. The steady state AHL dose responses of the two circuits are shown in Fig. 6a. While the nominal output values were comparable in the regulated and

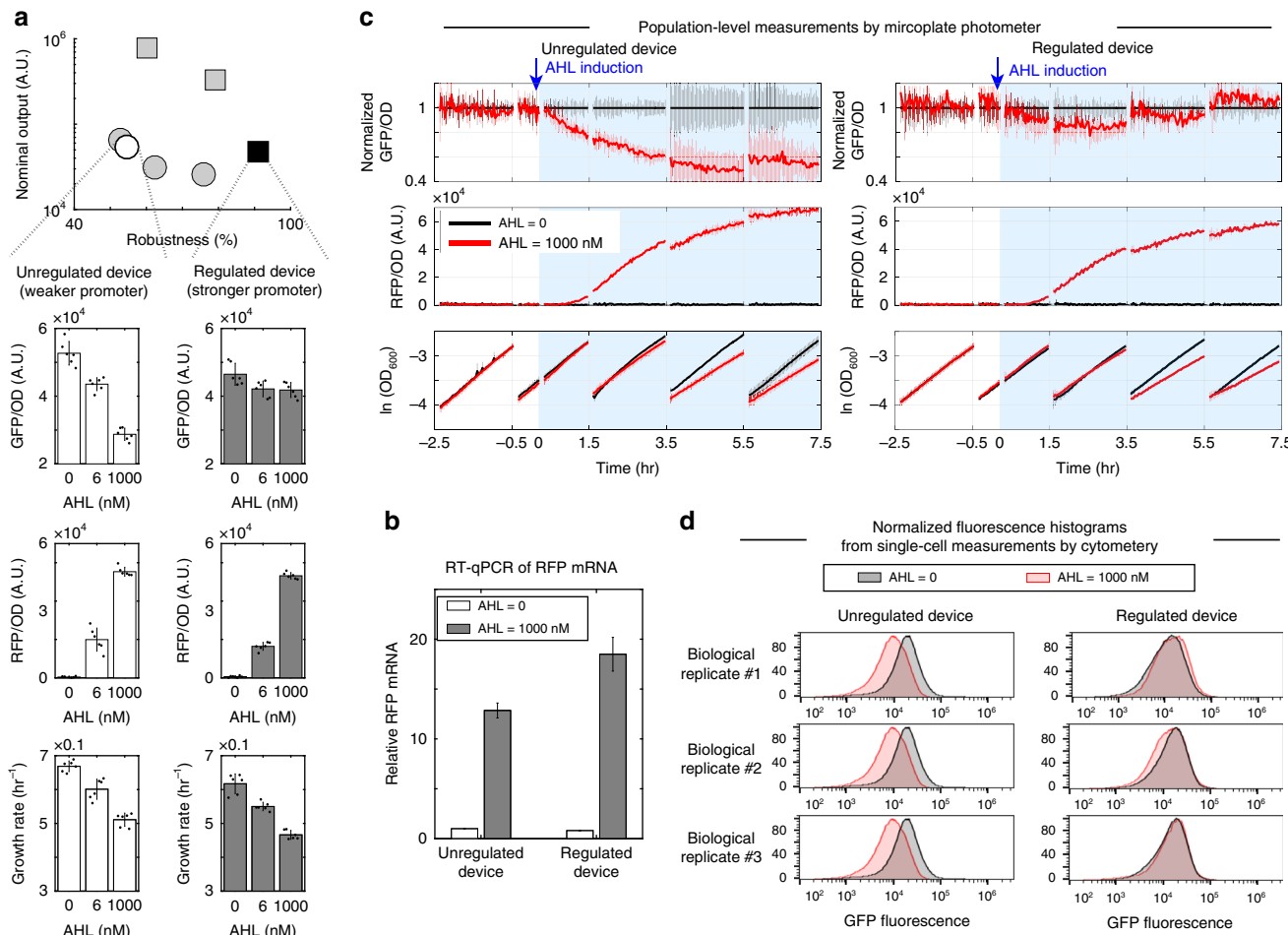

**Fig. 6** A regulated TX device is more robust than an unregulated device with comparable output level. **a** AHL dose responses of the regulated (black square in top panel) and unregulated (white circle in top panel) TX devices. The regulated device uses a stronger Ec-TTL-P109 promoter and the unregulated device uses a weaker BBa_J23116 promoter. For both circuits, the RBS TIRs of ECF32 are 6474, corresponding to high feedback gain. The robustness vs. nominal output chart of Fig. 4b is reported here to facilitate comparison. Error bars represent standard deviation from six replicates, including three biological replicates each having two technical replicates. Specific value of an independent experiment is presented as a black dot. **b** The mRNA level of mRFP1 gene was quantified by RT-qPCR and normalized to the reference gene cysG (see Method for details). Data represent mean values (±standard deviation). **c** Temporal responses of the unregulated and the regulated devices were monitored in parallel by microplate photometer. Cells were first grown in multiple batches in the absence of AHL for 8 h till they reach steady state GFP expression. Each batch was diluted every 2 h to maintain exponential cell growth. The resource competitor was then induced at $t = 0$ with the indicated AHL concentrations. Sampling time interval of microplate photometer was 2 min. Mean GFP per OD values in arbitrary unit (A.U.) are normalized to those of samples without AHL induction to reflect relative change in GFP expression. Error bars represent standard deviation from three biological replicates. Temporal data without normalization can be found in Supplementary Fig. 9. **d** GFP fluorescence histograms (normalized to the statistical mode) of three biological replicates measured by flow cytometry 6 h after AHL induction. Respective scatter plots of both RFP and GFP and their detailed statistics can be found in Supplementary Note 6. Data in **a–d** were collected from three independent experiments. Source data are provided as a Source Data file.Source Data

unregulated devices ($\sim 4.5 \times 10^4$ for the regulated TX device and $\sim 5.5 \times 10^4$ for the unregulated TX device), the robustness of the unregulated device was only 50% in contrast to a high robustness of nearly 95% for the regulated device (Fig. 6a). The resource competitors in both circuits produced similar levels of RFP and imparted comparable growth retardation, indicating that comparable disturbances were applied to the regulated and unregulated devices (Fig. 6a). This is further supported by RT-qPCR measurements, which confirm that when the resource competitor was fully activated, RFP mRNA levels in cells bearing the regulated devices were no less than the RFP mRNA levels found in cells bearing the unregulated devices (Fig. 6b).

In Fig. 6c, we report the dynamic responses of the two circuits when subject to a step disturbance input (i.e., AHL induction).

Consistent with the simulations in Fig. 3b, the unregulated TX device's output dropped after AHL induction and did not recover. By contrast, the output of the regulated TX device was nearly unaffected by the applied disturbance. The similar growth profiles of the regulated and unregulated devices, reflected in the values of optical density (OD) at 600 in Fig. 6c, show that the sRNA-mediated feedback did not unfavorably affect cell growth. With reference to Fig. 6d, we did not observe any appreciable difference in gene expression noise between the regulated and unregulated TX devices, consistent with experimental results by others[19] (see Supplementary Note 6 for detailed statistics). Altogether, the results in Fig. 6 demonstrate that the sRNA-mediated quasi-integral control is an effective approach to insulate the behavior of a TX device from variations in ribosome availability.

## Discussion

Modular design of genetic circuits relies on the assumption that the i/o behavior of TX devices is not affected by surrounding systems. This allows creation of increasingly sophisticated systems, whose behavior can be conveniently predicted by that of the composing devices characterized in isolation. While substantial progress has been made toward making the i/o behavior of a device independent of its context[3,4,6], a TX device's response can still be severely affected by operation of other devices through resource sharing[7–13]. This impedes a modular approach to designing complicated systems, hampering progress in synthetic biology. Here, we have proposed a solution to this problem, which allows the output response of a TX device in bacteria to adapt to changes in availability of ribosomes imparted by other devices. Our solution is based on embedding a post-TX feedback controller within a TX device, which treats changes in ribosomes concentration as a disturbance. The controller reaches near perfect adaptation by implementing the quasi-integral control strategy[26] through sRNA-mediated mRNA silencing. In a quasi-integral control scheme, perfect adaptation can be asymptotically reached as a feedback gain parameter is progressively increased. Here, we have demonstrated through combined theoretical and experimental studies that this parameter can be easily tuned through the ECF sigma factor's RBS strength (Fig. 5). This leads to a practical way to construct a high-gain regulated TX device that is robust to variations in ribosome availability and therefore amenable to modular circuit construction. Due to the post-TX nature of the controller, our design cannot attenuate the effects of perturbations or noise on transcription. To achieve adaptation to ribosome variability and robustness to perturbations affecting transcription may require the combination of previously proposed solutions to the latter problem[6,46] with our design.

Chemical reactions with similar structure to sRNA-mediated silencing have been proposed as a way to implement integral control under the assumption that molecule decay can be neglected[30,47]. While this assumption is reasonable in slow-growing cells and with controller species with negligible degradation, our experimental results suggest that this may not be the case for RNA species in *Escherichia coli*. In fact, under the assumption of negligible decay, perfect adaptation should be achieved regardless of the feedback gain. By contrast, our controller reaches perfect adaptation only for high feedback gain (Fig. 5). We have characterized the promoter pECF32 and found that it is not strong (Supplementary Note 1). This is an important aspect of the design as a strong pECF promoter combined with high feedback gain may lead to the pECF promoter to saturate, breaking the feedback loop (Supplementary Note 10).

In contrast to other efforts aimed to regulate gene expression by modulating its TX rate[24,48], our control design is post-TX and hence completely orthogonal to TX regulation. It thus enables seamless i/o composition of TX devices, consistent with a modular design concept. Our controller's constituent biomolecular parts, specifically the library of synthetic sRNAs and ECF sigma factors, are expandable and interchangeable[25,35]. Further exploration of these libraries can potentially lead to additional available parts that enable a generalizable and scalable solution to engineer modular genetic circuits with increased size and complexity.

While our work takes a decentralized control approach, wherein feedback controllers are embedded in TX devices, other researches have focused on centralized control approaches, where solutions globally manipulate resource pool size and/or allocation[34,49,50]. However, depending on the genetic circuit's topology, more abundant ribosomes can actually increase the relative strength of unintended interactions due to resource sharing (Supplementary Note 12). Therefore, what solution to use may largely depend on the application, with a decentralized solution being especially promising in applications involving signal transduction and logic computation, where accuracy and precision, rather than high yield, are the main design considerations. For future large integrated genetic circuits, a combination of both centralized and decentralized solutions may be optimal.

Homeostasis and adaptation has long been argued as a key property for survival of living organisms in highly uncertain and variable environments[51]. Several sRNA feedback motifs similar to ours have been identified in natural systems to regulate key physiological responses. For example, negative feedback regulation through sRNA is involved in extracytoplasmic stress response[52], iron homeostasis[53–55], quorum-sensing[56,57], and sugar metabolism[58,59]. Given our results, instances of such a feedback motif in natural biological pathways may hint to a previously under-appreciated ability of such pathways to withstand perturbations.

## Methods

**Strain and growth medium.** *E. coli* DIAL strain JTK-160J[60] was used in all constructions and experiments in this work. The growth medium was M9610 medium supplemented with 0.4% glucose, 0.2% casamino acids, and 1 mM thiamine hydrochloride. Appropriate antibiotics were added according to the selection marker of a genetic circuit. Final concentration of ampicillin or spectinomycin was 100 μg mL$^{-1}$. M9610 minimal medium was modified from M9 minimal medium to make phosphate buffer pH value to be 6 and its buffer capacity to be 10-fold of that of M9 minimal medium according to the Henderson–Hasselbalch equation. The recipe of M9610 minimal medium is Na$_2$HPO$_4$*7H$_2$O 33.36 mM, KH$_2$PO$_4$ 220.40 mM, NaCl 8.56 mM, and NH$_4$Cl 18.69 mM. The pH 6 and 30 °C growth conditions are intended to lower degradation rate of LuxR's effector AHL[61].

**Genetic circuit construction.** The genetic circuit construction was based on Gibson assembly[62]. DNA fragments to be assembled were amplified by PCR using Phusion High-Fidelity PCR Master Mix with GC Buffer (NEB, M0532S), purified with gel electrophoresis and Zymoclean Gel DNA Recovery Kit (Zymo Research, D4002), quantified with the nanophotometer (Implen, P330), and assembled with Gibson assembly protocol using NEBuilder HiFi DNA Assembly Master Mix (NEB, E2621S). Assembled DNA was transformed into competent cells prepared by the CCMB80 buffer (TekNova, C3132). Plasmid DNA was prepared by the plasmid miniprep-classic kit (Zymo Research, D4015). DNA sequencing used Quintarabio DNA basic sequencing service. The pVRa32_1122 and pVRb32_1122 plasmids for cloning ECF32 and its cognate promoter pECF32 were purchased from Addgene (ID 49689 and 49722, respectively). sRNA-A was synthesized in a gBlock. Primers and gBlocks were obtained from Integrated DNA Technologies. The list of primers and constructs can be found in Supplementary Tables 1–4 and Supplementary Figs. 1–3.

**Microplate photometer.** Overnight culture was prepared by inoculating a −80 °C glycerol stock in 800 μL growth medium per well in a 24-well plate (Falcon, 351147) and grew at 30 °C, 250 rpm in a horizontal orbiting shaker for 7 h. Overnight culture was first diluted to an initial OD at 600 nm (OD$_{600nm}$) of 0.02 in 200 μL growth medium per well in a 96-well plate (Falcon, 351172) and grew for 2–3 h to ensure exponential growth before induction. The 96-well plate was incubated at 30 °C in a Synergy MX (Biotek, Winooski, VT) microplate reader in static condition and was shaken at a fast speed for 3 s right before OD and fluorescence measurements. Sampling interval was 5 min unless stated otherwise in figure captions. Excitation and emission wavelengths to monitor GFP fluorescence are 485 and 530 nm, respectively, and those to monitor RFP fluorescence are 584 and 619 nm, respectively. To ensure exponential growth, cell culture was diluted every 2 h to OD of 0.02 as one batch. Multiple batches were used to ensure cell growth remains in exponential phase and gene expression reaches steady state. Growth rates were computed from the last batch of each experiment.

**Flow cytometry.** Single-cell fluorescence data of *E. coli* cells after 6-h AHL induction were measured by the Accuri C6 flow cytometer (Becton Dickinson, Special Order 2B2LYG RUO System, 656035). The optical system is equipped with 488 and 552-nm lasers to excite GFP and RFP, and employs the 525/50- and 610/25-nm band-pass filters to detect the emission of GFP and RFP, respectively. The fluid system sets the flow rate to 66 μL min$^{-1}$ and the core size to 22 μm. The detection threshold was set as 10,000 on FSC-H channel. Singlet events were gated in a FSC-A vs. FSC-H plot (Supplementary Fig. 11). At least 120,000 singlet events were collected for data analysis with the FlowJo v10 (FlowJo, LLC) and the CFlowPlus v1.0.264.14 (Becton Dickinson) softwares.

**RT-qPCR**. At the end of the last batch of an experiment ($OD_{600nm}$ ~0.05), 200 μL *E. coli* cells were lysed with 1150 μL DNA/RNA Shield 2× concentrate solution (Zymo Research R1200-125). Total RNA was extracted with Quick-RNA Microprep Kit (Zymo Research, R1051) and quantified with the nanophotometer (Implen, P330). The Script Flex cDNA Synthesis Kit (Quantabio, Cat. No. 95049-100) was used to synthesize first-strand cDNA with a mixed primer strategy (random primer plus Oligo dT) for the reference *cysG* gene[63] and with a gene-specific primer for the target *mRFP1* gene. Quantitative PCR (qPCR) reactions were prepared in 10 μL per reaction with Luna Universal qPCR Master Mix kit (New England Biolabs, M3003L) based on SYBR Green I dye by following the kit's manual and were amplified in a 384-well plate (Axygen, PCR-384-LC480-W) in the LightCycler480 (Roche). The ΔΔ $C_q$ method was used for qPCR data analysis from three biological replicates, each with three technical replicates. The RT-qPCR primers are listed in Supplementary Note 3.

**Mathematical modeling and simulation**. Derivation of the ODE model in Eqs. (2)–(4) is detailed in Supplementary Note 7. Numerical simulations are performed using MATLAB R2015b (The MathWotks, Inc., Natick, MA, USA) with variable step ODE solver ode23s. Simulation parameters are listed in Supplementary Table 9 and their choices are reasoned in Supplementary Note 13.

**Reporting summary**. Further information on experimental design is available in the Nature Research Reporting Summary linked to this article.

**Code availability**. Custom codes used for simulations in this research are provided as a Supplementary Software file.

## Data availability

Simulation and fluorescence data generated or analyzed during this study are included in the paper and its Supplementary Information files. Essential DNA sequences are provided in Supplementary Note 2. Full DNA sequences are available on Addgene (#120890-120901). A reporting summary for this Article is available as a Supplementary Information file. The source data underlying Figs. 3–6, Supplementary Figs. 5–10, Supplementary Figs. 15–18, and Supplementary Table 5 are provided as a Source Data file.

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

## Acknowledgments

This work was supported in part by AFOSR grant FA9550-14-1-0060 and NSF MCB Award number 1840257. Use of LightCycler480 in the RT-qPCR experiment was supported by the Koch Institute Support (core) Grant P30-CA14051 from the National Cancer Institute.

## Author contributions

D.D.V. designed the research. H.-H.H. created genetic constructs. Y.Q. developed mathematical models and directed the parameter tuning in the constructs. Y.Q. and H.-H.H. performed the experiments. All authors interpreted the data and wrote the manuscript.

## Additional information

**Competing interests:** The authors declare no competing interests.

