## [Peer Review File · Nature Communications]

Reviewers' Comments:

Reviewer #1:

Remarks to the Author:

When trying to engineer logic into genetic systems, especially when implementing complex synthetic networks, the burden imposed on the cell by additional heterologous expression, or the network itself, can greatly impair correct function. The authors present an elegant solution for the problem of synthetic device components being inconsistently expressed under varying levels of ribosome availability. The data reported shows that the device incorporates a regulatory RNA to form a negative feedback loop and maintain reporter output within impressive parameters when over-expression of a separate synthetic device is induced. Whilst the experimental data shown is impressive, there are several examples of key bits of data not being reported that would fully describe the dynamics of the system.

Major Comments

The use of RFP output as a reporter for TX device 2 ribosomal demand brings up a few questions that don't seem to be fully addressed in the manuscript. Is RFP output directly proportional to transcript level, i.e. the number of RBSs present in the cell, and are non-linear changes in RFP expression due to ribosome-depletion burden at high induction levels taken into account? This may need some transcriptomic analysis to establish the relationship. With the information currently in the manuscript I'm not convinced that RFP can be assumed to be a good proxy for ribosome depletion.

Reservations on the RFP proxy for ribosome availability aside, I can't find any experimentally generated data on the RFP output during the device characterisation. It seems strange to introduce it as a concept during the modelling and then not report the actual output. Rather than just inducer levels, RFP levels during experimental characterisation should be reported to properly evaluate how the system responds to ribosomal burden.

The comparison of GFP levels between regulated and unregulated devices (fig 4, S6) brings up a few questions. GFP outputs in the regulated devices are much lower than in the unregulated devices, by around 5-fold in a lot of cases, even at higher induction levels. This makes sense in that even at high induction levels, the regulated device is likely subject to post-transcriptional regulation from sRNA-A. This does, however, lead to a situation where consistency of expression at a low level (regulated) is being directly compared to consistency of expression at a higher level (unregulated). As it stands, GFP expression in the regulated device is not as high as expression in the unregulated device even at the highest induction levels. As the higher expression of GFP/ECF32 in the unregulated device is itself likely to be contributing to cellular burden more than in the regulated device, I don't think that this is a good control. To truly assess the benefit of the regulated vs unregulated device, an additional control in which GFP/ECF3 levels are in the same range in the unregulated device as in the regulated device should be provided. This could be through upregulation of GFP/ECF32 expression in the regulated device and/or minimal constitutive expression of the sRNA-A transcript in the unregulated control to bring expression levels down to the same levels as in the regulated device.

The difficulty in comparing GFP output levels between the regulated and unregulated devices is compounded by the lack of fluorescence data that has not been normalised to OD600 or the OD600 values for the fig4/S6 time-courses. I'd expect that the regulated and unregulated devices would have differing effects on overall cell fitness and, given that the purpose of these devices is to function in a living cell, the impact on OD600 would be of interest to anyone planning on using the device. A marked difference in growth between regulated and unregulated systems could also be affecting the normalised data given. The OD600 values for fig4/S6 should be given as a supplementary figure to allow proper evaluation.

Minor Comments

The statement in paragraph 1 of the introduction that “one has to re-design all circuits components every time a new component is added to a system” is very sweeping and I’m not sure is true in all cases e.g. where heterologous expression levels are low and not causing excess burden on the cell. The statement should probably be toned down to describe the situation as something that can happen commonly rather than being true in every instance.

In describing the results, the authors could be clearer in the separation of theoretically and experimentally derived data. I’m assuming that the data in figure 1B/D is a model-based prediction but I’ve not spotted this being stated in either the figure legend or the results text.

In the “Design of a sRNA-mediated post-TX controller” results section, I’m struggling to understand the concept of ECF32 being “regarded as a sensor that senses GFP concentration”. My understanding is that in this situation, GFP is both transcriptionally and translationally coupled to ECF32 and is in effect acting as a reporter of ECF32, and is used as such for the characterisation experiments. Postulating that the dynamics between GFP and ECF32 are similar to a target/sensor interaction seems a large conceptual leap to ECF32 being a sensor. The analogy should be toned down in the text or experimentally investigated.

In figure legends, the authors should clarify whether experimental replicates are technical or biological replicates. If biological replicates, standard deviation might be a more appropriate measure of error, if technical replicates, additional biological replicates would be required.

In the discussion, the authors state that the “...ECF sigma factors, are highly expandable and interchangeable” but when evaluating sigma factors in figure S5, two of the three sigma factors characterised showed a large impact on growth and the relationship between GFP output and aTc induction was not as desired. A demonstration that additional sigma factors from the library displayed the required dynamics would be needed before claiming that the system is highly expandable and interchangeable without appropriate caveats.

In the “Microplate photometer” methods section, were microplates shaking or static during time-course incubation in the plate-reader?

The “Multiplex turbidostat and automatic flow cytometry” methods section contains several grammatical errors. Fluorescence data was analysed by arithmetic mean, was the data normally distributed?

In figure S5, error bars are not shown for clearer data presentation. An additional figure should be included with error bars to assess the consistency of the data.

Reviewer #2:

Remarks to the Author:

In this manuscript the authors develop a synthetic small RNA that binds to and induces degradation of the target gene’s mRNA. This sRNA is then placed under the control of ECF sigma factor that senses the same gene’s product. The end result is a quasi-integral controller, in which the sRNA level approximates the integral of the recent history of the gene of interest and feeds back to activate or deactivate translation if the gene expression is too low or too high, respectively. The authors demonstrate that without this circuit, synthetic gene modules of interest in E coli are substantially depressed when other highly expressed genes outcompete for ribosome resources. With the quasi-integral controller, the circuit expression is relatively robust to changes in the competing species.

Overall, the manuscript is well written, and the conclusions of the study are supported by the results as presented. The results and methods appear to be novel, yet they simple and focused enough that they should be accessible to a broad audience in synthetic biology.

Comments:

1) Although it is clear that the proposed circuit does a good job to control the robustness of the population mean to the competing mRNA, I am very curious to see how that affects expression at the single-cell level. The authors note that they used flow cytometry to measure expression of both GFP and RFP at each point in time during response. This should have produced single-cell marginal and joint distributions for the two colors. The authors should present (at least in the SI) some of these scatter plots and explore how variability changes between the controlled and uncontrolled constructs. For example, does the relative amount of noise of the regulated output change compared to the unregulated construct? Do the correlations between GFP and RFP have a qualitative difference between the two constructs (e.g., uncorrelated in the regulated and anti-correlated in the regulated structure)? If clear differences are observed, can they be explained (at least qualitatively) by an extension of the current model to include stochasticity?

2) How sensitive are the controlled and uncontrolled models to variation in copy numbers for the different plasmids? One could imagine that robustness to ribosome availability has been achieved at the cost of greater sensitivity to plasmid copy levels.

3) It would be nice to see the method and circuit in work for a more complex situation where the GFP expression could be tuned at different levels (e.g., through aTc as in fig. 3) but at multiple different levels of competition (e.g., under different AHL induction levels). In other words, can the system faithfully track changes in aTc even within different or fluctuating AHL environments?

4) Can the authors fit their model in Figure 2 to the data in Fig. 4? This would be a much better way to compare the model and data.

Minor comments:

1) Page 4. (Typo) "Therefore, with reference to Figure 1C, ECF32 can be regarded as a sensor that senses GFP concentration to ACTUATE sRNA transcription."

2) Page 6. " θ is the degradation rate constant of the mRNA-sRNA complexes". This doesn't sound exactly right. I would think that θ is the binding rate of the mRNA and sRNA and either the degradation rate of the complex is assumed to be infinite or the unbinding rate is assumed to be zero.

3) The SI material has a very large number of typos. Please check this carefully.

A quasi-integral controller for adaptation of genetic modules to variable ribosome demand

Response to Reviewers

We appreciate the reviewers' detailed and constructive comments on our paper. We have incorporated them into the new manuscript. Detailed responses to the reviewers' comments can be found below, with the reviewers' comments itemized in black, and our responses in blue. We hope the reviewers are satisfied with our revision, and thank them for helping us improve the quality of this paper.

Response to Reviewer 1

When trying to engineer logic into genetic systems, especially when implementing complex synthetic networks, the burden imposed on the cell by additional heterologous expression, or the network itself, can greatly impair correct function. The authors present an elegant solution for the problem of synthetic device components being inconsistently expressed under varying levels of ribosome availability. The data reported shows that the device incorporates a regulatory RNA to form a negative feedback loop and maintain reporter output within impressive parameters when over-expression of a separate synthetic device is induced. Whilst the experimental data shown is impressive, there are several examples of key bits of data not being reported that would fully describe the dynamics of the system.

Major comments:

1. The use of RFP output as a reporter for TX device 2 ribosomal demand brings up a few questions that don't seem to be fully addressed in the manuscript. Is RFP output directly proportional to transcript level, i.e. the number of RBSs present in the cell, and are non-linear changes in RFP expression due to ribosome-depletion burden at high induction levels taken into account? This may need some transcriptomic analysis to establish the relationship. With the information currently in the manuscript I'm not convinced that RFP can be assumed to be a good proxy for ribosome depletion.

Response: We thank the reviewer for this comment. We have performed RT-qPCR experiments to confirm that the RFP mRNA levels for the regulated device is not smaller than that of the unregulated device upon AHL induction (see Figure 6B). In addition, we report comparable RFP protein levels for both devices in Figure 5C and 6C of the revised manuscript. These data indicate that the added demand for ribosomes upon AHL induction, dictated by the level of RFP mRNA, is not smaller in the regulated TX device.

2. Reservations on the RFP proxy for ribosome availability aside, I can't find any experimentally generated data on the RFP output during the device characterisation. It seems strange to introduce it as a concept during the modelling and then not report the actual output. Rather than just inducer levels, RFP levels during experimental characterisation should be reported

to properly evaluate how the system responds to ribosomal burden.

Response: We have provided experimental RFP data for all regulated and unregulated TX devices in Figures 5C, 6C (main text) and Figures S7-S12 (SI).

3. The comparison of GFP levels between regulated and unregulated devices (fig 4, S6) brings up a few questions. GFP outputs in the regulated devices are much lower than in the unregulated devices, by around 5-fold in a lot of cases, even at higher induction levels. This makes sense in that even at high induction levels, the regulated device is likely subject to post-transcriptional regulation from sRNA-A. This does, however, lead to a situation where consistency of expression at a low level (regulated) is being directly compared to consistency of expression at a higher level (unregulated). As it stands, GFP expression in the regulated device is not as high as expression in the unregulated device even at the highest induction levels. As the higher expression of GFP/ECF32 in the unregulated device is itself likely to be contributing to cellular burden more than in the regulated device, I don't think that this is a good control. To truly assess the benefit of the regulated vs unregulated device, an additional control in which GFP/ECF32 levels are in the same range in the unregulated device as in the regulated device should be provided. This could be through upregulation of GFP/ECF32 expression in the regulated device and/or minimal constitutive expression of the sRNA-A transcript in the unregulated control to bring expression levels down to the same levels as in the regulated device.

Response: We appreciate and followed the reviewer's helpful suggestion. In the revised manuscript, we compare a new regulated TX device driven by a stronger constitutive promoter and a new unregulated TX device driven by a weaker constitutive promoter. As shown in Figure 6, the two devices produce similar levels of GFP in the absence of AHL induction (with level 4.7×10^4 GFP/OD for regulated and level 5.3×10^4 GFP/OD for unregulated). However, while the regulated TX device was nearly unaffected by the resource competitor, GFP expression by the unregulated TX device was reduced by nearly 50% upon competitor activation. Compared to the regulated and the unregulated devices we reported in the old manuscript, which both used the BBa_J23110 promoter, the GFP expression from the two new devices is lower than expression found in the unregulated device in the old manuscript ($\sim 5 \times 10^4$ GFP/OD for both new devices compared to $\sim 8 \times 10^4$ GFP/OD for the old unregulated device) and much higher than GFP expression in the regulated device in the old manuscript ($\sim 1 \times 10^4$ GFP/OD). We hope these new experiments better demonstrate the benefit of the post-TX controller.

4. The difficulty in comparing GFP output levels between the regulated and unregulated devices is compounded by the lack of fluorescence data that has not been normalised to OD600 or the OD600 values for the fig4/S6 time-courses. I'd expect that the regulated and unregulated devices would have differing effects on overall cell fitness and, given that the purpose of these devices is to function in a living cell, the impact on OD600 would be of interest to anyone planning on using the device. A marked difference in growth between regulated and unregulated systems could also be affecting the normalised data given. The OD600 values for fig4/S6 should be given as a supplementary figure to allow proper evaluation.

Response: Thank you for this suggestion. We have now included OD₆₀₀ values as well as growth rates for all TX devices in the revised manuscript (see Figures 5-6 and Figure S7-S10). For the regulated and unregulated devices in Figure 6, which have similar GFP expression levels, growth rates were comparable without competitor activation (Figure 6A, C). When the resource competitor was activated, similar extent of growth retardation were observed for the two devices (Figure 6C). This observation is consistent with RT-qPCR and RFP measurements in Figure 6A-C, which indicate similar levels of ribosome perturbation were applied to the two devices by the resource competitor.

Minor comments:

1. The statement in paragraph 1 of the introduction that “one has to re-design all circuits components every time a new component is added to a system” is very sweeping and Im not sure is true in all cases e.g. where heterologous expression levels are low and not causing excess burden on the cell. The statement should probably be toned down to describe the situation as something that can happen commonly rather than being true in every instance.

Response: Thank you. We have rephrased this sentence to the following:

“This frequently leads to a combinatorial design problem where one has to re-design a circuit’s components when a new component is added.”

2. In describing the results, the authors could be clearer in the separation of theoretically and experimentally derived data. Im assuming that the data in figure 1B/D is a model-based prediction but Ive not spotted this being stated in either the figure legend or the results text.

Response: Thank you. Figures 1B/D are cartoons drawn to illustrate the modularity concepts and are not based on theoretical or experimental results. This statement has been explicitly included in the caption of Figure 1. Relevant simulation and experimental results are shown in Figure 2-5.

3. In the “Design of a sRNA-mediated post-TX controller” results section, Im struggling to understand the concept of ECF32 being “regarded as a sensor that senses GFP concentration”. My understanding is that in this situation, GFP is both transcriptionally and translationally coupled to ECF32 and is in effect acting as a reporter of ECF32, and is used as such for the characterisation experiments. Postulating that the dynamics between GFP and ECF32 are similar to a target/sensor interaction seems a large conceptual leap to ECF32 being a sensor. The analogy should be toned down in the text or experimentally investigated.

Response: We appreciate the reviewer’s comment and realize that this point was not clearly explained in the main text. In general, the ECF concentration level in time is not necessarily proportional to the GFP concentration since GFP and ECF are only transcriptionally but not translationally coupled. This is however the case at steady state if both proteins are stable, that is, when their decay rates are only dictated by dilution and hence approximately equal. Under this circumstance, we can mathematically show (SI equations (S13)-(S14)) that the ratio between the steady state protein concentrations is given by the ration between the RBS strengths. For this to be theoretically true also during the temporal response, it is necessary that the initial concentrations of ECF and GFP satisfy this proportionality relationship. This is the case if, for example, we start the experiment from zero concentrations or from a steady state situation, which are the conditions under which we perform our experiments. This is more clearly explained in SI Section B1.

We therefore agree with the reviewer that the analogy to an engineering sensor should not be carried too strictly and that ECF is not a true sensor of GFP. To address this, we slightly changed the block diagram in Figure 1A,C, to be more truthful to the quantities that are physically sensed by the controller. Specifically, the ECF TL rate is proportional to the GFP TL rate and hence the input to the TX controller is physically the GFP’s TL rate and not quite its concentration (see also SI Section B1 equation (S9)). This is further reflected in the more detailed block diagram that we added as Figure 2B, in which the feedback gain k is introduced. We hope that this clarifies the confusion.

4. In figure legends, the authors should clarify whether experimental replicates are technical or biological replicates. If biological replicates, standard deviation might be a more appropriate

measure of error, if technical replicates, additional biological replicates would be required.

Response: Thank you for this suggestion. We have included the number of biological and technical replicates for all experiments in the Figure’s captions. Error bars now indicate standard deviations.

5. In the discussion, the authors state that the “...ECF sigma factors, are highly expandable and interchangeable” but when evaluating sigma factors in figure S5, two of the three sigma factors characterised showed a large impact on growth and the relationship between GFP output and aTc induction was not as desired. A demonstration that additional sigma factors from the library displayed the required dynamics would be needed before claiming that the system is highly expandable and interchangeable without appropriate caveats.

Response: Thank you for your comment. We have re-written this paragraph to tone down our claim:

“Our controller’s constituent biomolecular parts, specifically the library of synthetic sRNAs and ECF sigma factors, are expandable and interchangeable [24,34]. Further exploration of these libraries can potentially lead to additional available parts that enable a generalizable and scalable solution to engineer modular genetic circuits with increased size and complexity. ”

6. In the “Microplate photometer” methods section, were microplates shaking or static during time-course incubation in the plate-reader?

Response: The microplate were static during incubation but were shaken for 3 seconds prior to every measurement. This has been clearly stated in the methods section of the revised manuscript.

7. The “Multiplex turbidostat and automatic flow cytometry” methods section contains several grammatical errors. Fluorescence data was analysed by arithmetic mean, was the data normally distributed?

Response: We have re-written the flow cytometry methods section. For the revised manuscript, we did not collect any data using the multiplex turbidostat and automatic cytometry, instead, cells were incubated in the microplate in the plate-reader and then measured manually using flow cytometry. The scatter plots and the histograms are now reported in Figure 6D and Figure S11-S12, where multiple statistics of the cell population, including mean, median and geometric mean, have been reported. Evaluation of these statistics show consistent difference between the regulated and the unregulated devices.

8. In figure S5, error bars are not shown for clearer data presentation. An additional figure should be included with error bars to assess the consistency of the data.

Response: We have revised Figure S5 to include the error bars.

Response to Reviewer 2

In this manuscript the authors develop a synthetic small RNA that binds to and induces degradation of the target genes mRNA. This sRNA is then placed under the control of ECF sigma factor that senses the same genes product. The end result is a quasi-integral controller, in which the sRNA level approximates the integral of the recent history of the gene of interest and feeds back to activate or deactivate translation if the gene expression is too low or too high, respectively. The authors demonstrate that without this circuit, synthetic gene modules of interest in E coli are substantially depressed when other highly expressed genes outcompete for ribosome resources. With the quasi-integral controller, the circuit expression is relatively robust to changes in the competing species.

Overall, the manuscript is well written, and the conclusions of the study are supported by the results as presented. The results and methods appear to be novel, yet they simple and focused enough that they should be accessible to a broad audience in synthetic biology.

Major comments:

1. Although it is clear that the proposed circuit does a good job to control the robustness of the population mean to the competing mRNA, I am very curious to see how that affects expression at the single-cell level. The authors note that they used flow cytometry to measure expression of both GFP and RFP at each point in time during response. This should have produced single-cell marginal and joint distributions for the two colors. The authors should present (at least in the SI) some of these scatter plots and explore how variability changes between the controlled and uncontrolled constructs. For example, does the relative amount of noise of the regulated output change compared to the unregulated construct? Do the correlations between GFP and RFP have a qualitative difference between the two constructs (e.g., uncorrelated in the regulated and anti-correlated in the unregulated structure)? If clear differences are observed, can they be explained (at least qualitatively) by an extension of the current model to include stochasticity?

Response: We appreciate the reviewer’s helpful comment. In the revised manuscript, we have included the scatter plots and histograms of GFP and RFP obtained from flow cytometry in Figure 6D and Figure S11-S12. We did not observe appreciable GFP variability difference in the regulated and the unregulated devices (see Figure S11-S12 and statistics summarized therein). This observation is consistent with recent experimental results by Kelly et al. [1], which specifically studied the stochastic properties of sRNA-mediated feedback.

For both the regulated and the unregulated devices, GFP and RFP were weakly positively correlated, with correlation coefficients between 0.2-0.5 (Tables S5-S6). At least the following two aspects of cell-cell variability may contribute to this weak positive correlation between GFP and RFP.

- GFP and RFP are on the same plasmid. Even in the presence of ribosome competition and/or sRNA-mediated feedback, expression of both GFP and RFP increase with plasmid copy number (see SI Section B1). Therefore, a stochastic increase (decrease) in plasmid copy number leads to simultaneous increases (decreases) in both GFP and RFP.
- Similarly, GFP and RFP use the same pool of cellular resources. As a consequence, a stochastic increase (decrease) in the total number of ribosomes leads to increases (decreases) in both GFP and RFP expression.

Consequently, cell-cell variability in plasmid copy number and total cellular resources both lead to positively correlated GFP and RFP expression. We think a more detailed study on the stochastic properties of these circuits requires additional experiments specifically designed to

[Redacted]

[Redacted]

[redacted]

Minor comments:

1. Page 4. (Typo) “Therefore, with reference to Figure 1C, ECF32 can be regarded as a sensor that senses GFP concentration to ACTUATE sRNA transcription.”

Response: Thank you. We have fixed this typo.

2. Page 6. “ θ is the degradation rate constant of the mRNA-sRNA complexes”. This doesnt sound exactly right. I would think that θ is the binding rate of the mRNA and sRNA and

either the degradation rate of the complex is assumed to be infinite or the unbinding rate is assumed to be zero.

Response: Thank you. We have modified the model in the main text to use two parameters λ and β to describe mRNA-sRNA interaction. Specifically, λ is the degradation rate of the mRNA-sRNA complex and β is the dissociation constant of mRNA and sRNA binding.

3. The SI material has a very large number of typos. Please check this carefully.

Response: Thank you. We have proofread the SI to eliminate the typos.

References

- [1] C. L. Kelly, A. W. K. Harris, H. Steel, E. J. Hancock, J. T. Heap, and A. Papachristodoulou. Synthetic negative feedback circuits using engineered small RNAs. *Nucleic. Acids. Res.*, 2018.

Reviewers' Comments:

Reviewer #1:

Remarks to the Author:

I am satisfied with the reviewers rebuttal and manuscript changes in response to my reviewing comments. I feel that all of my queries and suggestions have been well addressed and am happy to recommend the manuscript for publication in its current form.

Benjamin Blount (Reviewer 1)

Reviewer #2:

Remarks to the Author:

The revised manuscript is much improved and makes a convincing argument that a quasi-integral controller, using sRNA to integrate gene expression set point errors and respond to control translation, can be an effective means to reduce sensitivity to fluctuations in resources due to competing processes.

The authors' responses to previous questions were satisfactory to me, and the additional data improves the presentation of the paper.

However, a minor question is raised by the newly provided cellular variability data:

The new Figures 6D, S11, and S12 clearly show that the introduced mechanisms reduce sensitivity of the mean GFP dynamics to the activation of RFP expression (i.e., the presumed ribosomal load). However, the shift due to the load appears to be minor when compared to other contributors to variability. For example, Fig S11 shows that a 100x increase of RFP causes a <50% decrease to average GFP expression, which is consistent with cited work [Refs 7, 11], but less than the 65-75% variation seen in the original population. If ribosomal competition were a dominant player in signal robustness, and if ribosomal usage were variable among a population, then you should see a reduction in cellular heterogeneity with the addition of feedback. However, the opposite occurs in that the GFP CV increases from 64-66% in Fig. S11 to 68-82% in Fig. S12, and the broadening of the distribution with the regulated system is visibly clear in Fig. 6.

The fact that the introduced feedback mechanism beautifully compensates for increased ribosomal loads (and presumably other translational or post translational effects) but does not significantly reduce natural variability suggests that natural variability is controlled by other pre-translational mechanisms (e.g., transcription or plasmid copy numbers) or measurement noise (e.g., noisy fluorescence detection). Perhaps I am missing something, but this seems to raise a question to the otherwise well-supported opening statement, "While many factors contribute to context-dependence of genetic modules, sharing limited gene expression resources has appeared as a major player into this problem." The authors should comment on this remaining variability and acknowledge that their work to address ribosomal load effects remains to be combined with future explorations to address other 'major players' that challenge modular design in synthetic biology.

A quasi-integral controller for adaptation of genetic modules to variable ribosome demand

Response to Reviewers

We appreciate the reviewers' constructive comments. We have updated the new manuscript to address the comment by Reviewer 2. We hope the reviewers are satisfied with our revision and thank them again for helping us improve the quality of this paper.

Response to Reviewer 1

I am satisfied with the reviewers rebuttal and manuscript changes in response to my reviewing comments. I feel that all of my queries and suggestions have been well addressed and am happy to recommend the manuscript for publication in its current form.

Response: Thank you.

Response to Reviewer 2

The revised manuscript is much improved and makes a convincing argument that a quasi-integral controller, using sRNA to integrate gene expression set point errors and respond to control translation, can be an effective means to reduce sensitivity to fluctuations in resources due to competing processes.

The authors responses to previous questions were satisfactory to me, and the additional data improves the presentation of the paper. However, a minor question is raised by the newly provided cellular variability data:

The new Figures 6D, S11, and S12 clearly show that the introduced mechanisms reduce sensitivity of the mean GFP dynamics to the activation of RFP expression (i.e., the presumed ribosomal load). However, the shift due to the load appears to be minor when compared to other contributors to variability. For example, Fig S11 shows that a 100x increase of RFP causes a <50% decrease to average GFP expression, which is consistent with cited work [Refs 7, 11], but less than the 65-75% variation seen in the original population. If ribosomal competition were a dominant player in signal robustness, and if ribosomal usage were variable among a population, then you should see a reduction in cellular heterogeneity with the addition of feedback. However, the opposite occurs in that the GFP CV increases from 64-66% in Fig. S11 to 68-82% in Fig. S12, and the broadening of the distribution with the regulated system is visibly clear in Fig. 6.

The fact that the introduced feedback mechanism beautifully compensates for increased ribosomal loads (and presumably other translational or post translational effects) but does not significantly reduce natural variability suggests that natural variability is controlled by other pre-translational mechanisms (e.g., transcription or plasmid copy numbers) or measurement noise (e.g., noisy fluorescence detection). Perhaps I am missing something, but this seems to raise a question to the otherwise well-supported opening statement, While many factors contribute to context-dependence of genetic modules, sharing limited gene expression resources has appeared as a major player into

this problem. The authors should comment on this remaining variability and acknowledge that their work to address ribosomal load effects remains to be combined with future explorations to address other major players that challenge modular design in synthetic biology.

Response: We agree with the reviewer that many factors contribute to context dependence in genetic circuits. This has been clearly stated in the Introduction. Our controller is designed to address unexpected couplings among genetic modules arising from ribosome competition, or more broadly speaking, disturbances occurring during the post-TX processes. The feedback is not intended to reduce cell-cell heterogeneity. In fact, as the reviewer suggested, due to its post-TX nature, the controller cannot reduce cell-cell heterogeneity arising from both intrinsic noise (i.e., natural randomness associated with biomolecular reactions) and extrinsic noise (e.g., variability in enzyme amount or plasmid copy number) in TX processes [1]. In addition, its ability to reduce intrinsic noise in post-TX processes still needs to be studied more carefully in a stochastic model. To clearly state the controller’s capabilities and limitations, we have added the following sentences in the first paragraph of the Discussion section as the reviewer suggested:

“Due to the post-TX nature of the controller, our design cannot attenuate the effects of perturbations or noise on transcription. To achieve adaptation to ribosome variability and robustness to perturbations affecting transcription may require the combination of previously proposed solutions to the latter problem [2, 3] with our design.”

References

- [1] P. S. Swain, M. B. Elowitz, and E. D. Siggia. Intrinsic and extrinsic contributions to stochasticity in gene expression. *Proc. Natl. Acad. Sci. U. S. A.*, 99(20):12795–12800, 2002.
- [2] Thomas H. Segall-Shapiro, Eduardo D. Sontag, and Christopher A. Voigt. Engineered promoters enable constant gene expression at any copy number in bacteria. *Nat. Biotechnol.*, 36(4):352–358, 2018.
- [3] Corentin Briat, Ankit Gupta, and Mustafa Khammash. Antithetic proportional-integral feedback for reduced variance and improved control performance of stochastic reaction networks. *J. R. Soc. Interface*, 15(143):20180079, 2018.